# Composites of Lignin-Based Biochar with BiOCl for Photocatalytic Water Treatment: RSM Studies for Process Optimization

**DOI:** 10.3390/nano13040735

**Published:** 2023-02-15

**Authors:** Amit Kumar Singh, Dimitrios A. Giannakoudakis, Michael Arkas, Konstantinos S. Triantafyllidis, Vaishakh Nair

**Affiliations:** 1Department of Chemical Engineering, National Institute of Technology Karnataka (NITK), Surathkal, Mangalore 575025, India; 2Laboratory of Chemical and Environmental Technology, Department of Chemistry, Aristotle University of Thessaloniki, 54124 Thessaloniki, Greece; 3Demokritos National Centre for Scientific Research, Institute of Nanoscience and Nanotechnology, 15310 Athens, Greece

**Keywords:** Biochar, BiOCl, photocatalysis, wastewater, azo dye remediation, lignin for material, RSM optimization

## Abstract

Textile effluents pose a massive threat to the aquatic environment, so, sustainable approaches for environmentally friendly multifunctional remediation methods degradation are still a challenge. In this study, composites consisting of bismuth oxyhalide nanoparticles, specifically bismuth oxychloride (BiOCl) nanoplatelets, and lignin-based biochar were synthesized following a one-step hydrolysis synthesis. The simultaneous photocatalytic and adsorptive remediation efficiency of the Biochar–BiOCl composites were studied for the removal of a benchmark azo anionic dye, methyl orange dye (MO). The influence of various parameters (such as catalyst dosage, initial dye concentration, and pH) on the photo-assisted removal was carried out and optimized using the Box–Behnken Design of RSM. The physicochemical properties of the nanomaterials were characterized by scanning electron microscopy, energy-dispersive X-ray spectroscopy, X-ray diffraction, thermogravimetric analysis, nitrogen sorption, and UV–Vis diffuse reflectance spectroscopy (DRS). The maximum dye removal was observed at a catalyst dosage of 1.39 g/L, an initial dye concentration of 41.8 mg/L, and a pH of 3.15. The experiment performed under optimized conditions resulted in 100% degradation of the MO after 60 min of light exposure. The incorporation of activated biochar had a positive impact on the photocatalytic performance of the BiOCl photocatalyst for removing the MO due to favorable changes in the surface morphology, optical absorption, and specific surface area and hence the dispersion of the photo-active nanoparticles leading to more photocatalytic active sites. This study is within the frames of the design and development of green-oriented nanomaterials of low cost for advanced (waste)water treatment applications.

## 1. Introduction

The extreme expansion of industrialization, urbanization, and the world’s population speed up the consumption of natural clean energy and causes severe environmental pollution [1,2,3]. Wastewater treatment and sustainable ways to manage environmental pollutants are among the most critical global challenges of the last few decades [4,5,6,7,8,9]. Due to their low biodegradability, high toxicity, high chemical stability, and complex cyclic structure, dyes and pigments seriously threaten aquatic ecosystems [10,11,12]. A significant amount of dye enters the environment through industrial effluents from the textile, food, petrochemical, and pharmaceutical industries [13,14]. Furthermore, this wastewater can harm public health and natural ecosystems due to its improper treatment. Therefore, an eco-friendly, sustainable, and cost-effective way to resolve this issue is necessary, bringing photocatalysis into perspective [15,16,17,18,19]. Photocatalysis allows using clean, safe, and naturally available energy with semiconductor photocatalysts for degrading organic pollutants, which is a significant advancement [20,21,22].

BiOCl is a semiconductor with several advantages such as a visible light response, high catalytic activity, controllable morphology, simple preparation process, and low environmental toxicity; hence it can be used as photocatalytic material [23]. BiOCl’s layered structure and internal electric field allow for its extensive application in degrading water pollutants, dyes [24,25], phenolic groups [26,27,28], and antibiotics [29]. However, some of its drawbacks are the large band gap of BiOCl (~3.4 eV) and its weak charge separation efficiency. Many researchers have demonstrated that constructing composite materials could enhance the photocatalytic performance of BiOCl [30,31,32,33,34,35,36]. Carbon materials have been reported to promote the separation of photogenerated electron-hole pairs, but they also suffer drawbacks such as complex synthesis procedures, high cost, etc. [37,38]. Therefore, developing a low-cost, abundant carbon material with better photocatalytic and optical properties than BiOCl is necessary to overcome these drawbacks.

Lignin is an essential cell wall constituent and the second most abundant polymer. Worldwide, about five million metric tons of lignin are produced yearly as a non-commercialized waste product; only two percent of total industrial lignin is valorized yearly [39]. Currently, lignin-based waste products are highly undervalued and are primarily used to produce energy. Kraft lignin is the prime by-product of the paper and pulp industry produced after the kraft process and is the most underutilized material. Employing process modifications and cost-effective methods to extract Kraft lignin provides a more efficient and value-added utilization of lignin. Therefore, lignin offers a novel, sustainable, eco-friendly, and cost-effective approach to biochar production.

Biochar is a low-cost, environmentally friendly, and sustainable carbonaceous material obtained from the pyrolysis of available waste biomass under limited oxygen conditions. Biochar has excellent potential for managing industrial waste biomass, thus decreasing the environmental pollution load. The use of different waste biomasses as biochar precursors is sustainable as well as economical. Waste biomass considered for biochar production includes paper mill waste, crop residues, food processing waste, municipal solid waste, forestry waste, and sewage sludge [40,41]. Biochar has tunable functional groups, a developed surface structure, chemical stability, and unique photoelectric properties favorable [42] for doping with photocatalysts that have received researchers’ attention. Biochar-supported photocatalysts have been used in novel, environmentally friendly, and effective methods for organic pollutants removal from wastewater [43,44].

The conventional way to evaluate photodegradation depends on changing an independent variable such as the pH, catalyst dosage, and initial concentration while keeping other parameters constant [45]. Consequently, the traditional method needed extra chemicals, time, and energy consumption to evaluate each variable. Therefore, to resolve the issue, RSM is used for process optimization and the design of experiments through different models such as the Box–Behnken design (BBD), central composite design (CCD), etc. RSM mainly establishes relations between independent variables and their responses using a minimum number of experiments [38]. The RSM technique is used in various applications such as in pharmaceuticals, chemical processes, microbiology, etc., to evaluate and optimize the interactive effect among the influencing factors [46,47].

In this study, we utilize chemically activated biochar prepared from lignin to fabricate BC-BiOCl composite photocatalysts via a one-step hydrolysis method. Herein, we degrade MO dye to evaluate the enhancement in photocatalytic performance of BiOCl after introducing activated biochar. Different characterization tests were performed to determine the composition, morphology, structure, and surface chemical properties of the as-prepared samples. The photocatalytic degradation of MO under different operating states, such as the initial dye concentration, catalyst dosage, and pH, was investigated. BBD of RSM is employed to optimize the experimental variables and evaluate the relation between operating factors.

## 2. Materials and Methods

### 2.1. Materials

Ethylene glycol (C_2_H_6_O_2_, assay 98%), potassium chloride (KCl, assay 99%), potassium hydroxide (KOH, assay 85%), and bismuth nitrate pentahydrate Bi(NO_3_)_3_.5H_2_O, assays 98%) were purchased from Molychem. Methyl orange (MO) from Spectrum reagents and chemicals, alkaline lignin from the Tokyo chemical industry, absolute ethanol (assay 99%) from Changshu Hongsheng fine chemicals, sodium hydroxide pellets (NaOH, assay 97%) from Isochem laboratories, and hydrochloric acid (HCl, assay 35%) were used as reagents. Distilled water was used to prepare solutions and the chemicals were used without alteration.

### 2.2. Synthesis of Biochar and BC-BiOCl Photocatalyst

Biochar was prepared using alkaline lignin as a raw material via a one-step preparation method [48]. Alkaline lignin and the activating agent potassium hydroxide (KOH) were taken in a 1:1 weight ratio and impregnated. The mixture was then subjected to carbonization in a muffle furnace at 700 °C for 60 min. After getting cooled, the generated activated biochar was washed withM HCl, warm water, and normal distilled water three times. The sample was then dried at 100 °C for 6 h and stored.

The BC-BiOCl photocatalysts were prepared via a one-step hydrolysis method [49]. At first, two mmol of bismuth nitrate pentahydrate Bi(NO_3_)_3_.5H_2_O was completely dispersed in 20 mL of ethylene glycol at room temperature. Then for different weight ratios, a particular amount of biochar was added to the Bi(NO_3_)_3_.5H_2_O solution. After that, in 20 mL of distilled water, 2 mmol of KCl was dissolved and added dropwise to the biochar and bismuth nitrate suspension. The mixture was stirred for 12 h at room temperature and filtered through filter paper. Then the filtered product was washed with pure ethanol and deionized water three times and dried at 60 °C. Samples with different weight ratios of biochar to BiOCl, namely 0%, 5%, 10%, 15%, and 20% were prepared and marked as BiOCl, 5% BC-BiOCl (5BCPC), 10% BC-BiOCl (10BCPC), 15% BC-BiOCl (15BCPC), and 20% BC-BiOCl (20BCPC), respectively.

### 2.3. Characterization of BiOCl-Biochar

The surface morphology of different samples was characterized using field emission scanning electron microscopy (FESEM; Carl Zeiss GeminiSEM 300, 2.00 kV). In addition, the elemental surface distribution was obtained using energy-dispersive X-ray spectroscopy (EDX; EDAX-Ametek, 0–10 KeV), with the focused elements being Bi, O, Cl, and C. Crystalline structures of the prepared samples were characterized using X-ray diffraction (XRD; Malvern PANalytical Empyrean 3rd Gen, Cu Kα wavelength = 1.54 Å, 45 kV, 40 mA) between 3° to 80° with a scan rate of 8° per minute. To determine the sample’s optical absorption and band gap energy, UV–Vis DRS was conducted. The synthesized samples’ absorbance and band gap energy were determined in the wavelength range of 300 to 800 nm. The thermal properties of the as-prepared samples were determined through Thermogravimetric Analysis (TGA) using the thermal analyzer TGA4000. At a heating rate of 10 °C/min, the samples were heated in the temperature range of 30 to 800 °C. The pHpzc was evaluated using the pH drift method by agitating solutions of 0.01 M NaCl solution with the pH varying from 2 to 12 containing 0.15 g of photocatalyst for 48 h. N2 adsorption–desorption studies were conducted to determine the adsorption capabilities of different samples using a Brunauer–Emmett–Teller (BET) analyzer (Autosorb IQ-XR-XR, Anton Paar, Austria). Before each adsorption measurement, the samples were degassed for 5 h under nitrogen at 80 °C.

### 2.4. Photocatalytic Degradation Studies

Solutions of MO were subjected to adsorption under dark conditions for 150 min to evaluate the adsorption capacity and kinetics of the synthesized photocatalyst. The general parameters for the preliminary study were 200 mL of 30 ppm concentrated MO solution at the unadjusted pH of 6.3 with 0.2 g of photocatalyst in a 300 mL crystallizing dish under 350 rpm stirring. The solution was exposed to a single UV-A (λ = 356 nm) lamp with 6 W power, fixed at a distance of 10 cm throughout the study. The best-performing photocatalyst was selected based on the preliminary experiments for further studies. Different parameters governing MO degradation, such as dye concentration (10–50 ppm), photocatalyst dosage (0.5–1.5 g/L), and pH (3–7), were investigated. Preceding the photocatalytic experiment, the MO solution with the photocatalyst was stirred in the dark for 30 min to achieve an equilibrium between the photocatalyst and dye solution. After 30 min of dark adsorption, the solution was irradiated under UV-A light for 2 h. After every 30 min, 2.5 mL of sample was collected using a 0.45 µm syringe filter, and the supernatant was analyzed using a UV–Vis spectrophotometer (Shimadzu UV 2600) at a maximum adsorption wavelength of 464 nm for the MO. To calculate the MO degradation efficiency the following formula was used:Degradation efficiency (%) = [(C_0_ − C_t_)/C_0_] *×* 100%(1)
where C_0_ is the initial dye concentration (ppm) and C_t_ is the dye concentration at a specified time interval t.

### 2.5. Experiment Design and Analysis

The Box–Behnken design (BBD) of RSM was employed to design the experiment and validate the model for MO degradation using the BC-BiOCl photocatalyst. The experiment design consisted of three factors and three levels for each factor. The catalyst dosage (g/L), dye concentration (ppm), and pH were selected as independent factors, and the MO degradation (%) was taken as the dependent factor for the RSM analysis. The selected independent factors were normalized at three different levels, i.e., high (+1), center (0), and low (−1). Furthermore, the range for these independent parameters was chosen based on preliminary studies. The BBD model analyzed the experimental data and fitted them in a quadratic equation of the form:(2)$$Y=+\sum_{i=1}^{n}a_{i}x_{i}+\sum_{i=1}^{n}a_{ii}x_{i2}+\sum_{i=1}^{n-1}\;\sum_{i=1}^{n}a_{ij}x_{i}x_{j}+Ɛ$$
where *Y* refers to the predicted response, $a_{0}$ is the coefficient constant, $a_{i}\;$ is the linear coefficient, $a_{ii}$ is a quadratic coefficient, $a_{ij}\;$ is the interaction coefficient among the factors, $x_{i}\;$ and $x_{j}$ are coded values of parameters, and Ɛ is the model error [50].

## 3. Results and Discussion

### 3.1. Characterization of the Pure Phases (BiOCl Nanoparticles and Lignin-Derived Activated Biochar)

The first task was to examine the physicochemical features of the received “pure” inorganic phase in order to conclude if the followed synthetic protocol led to nanostructured BiOCl. From the scanning electron microscopy (SEM) and high resolution, the SEM (HR-SEM) it can be concluded that the obtained material consisted of uniform platelet/disk-like shaped nanoparticles (Figure 1a) with a thickness of 15–25 nm and diameter in the range from 55 to 160 nm, with an average of 95 nm. The XRD pattern of the pure BiOCl-NPs fit absolutely with the standard tetragonal BiOCl data (JCPDS No. 85–861), with the characteristic reflection at (011), (110), and (012) (Figure 2) to be the predominant ones. As it was expected, the values of the textural parameters as estimated from the N_2_ sorption tests were not so high, but in accordance with analogues reported in the literature for the same class of materials. The BiOCl sample presented a specific surface (S_BET_) of 19 m^2^/g and a non-porous nature (total pore volume of 0.037 cm^3^/g), with the formed “pores” to be as a result of the formed inter-particles, inter-clusters, or/and inter-aggregate cages/spaces within the nanoparticles [51].

Regarding the pure alkaline lignin-derived biochar, it had a uniform structure (Figure 1b) of a not-densely packed nature, where uneven cages and cracks with openings at the macro-scale exist. The nitrogen sorption isotherm (Figure 3) revealed that the material had a broad pore size distribution, but was predominately micro-porous since the isotherm can be designated as *Type I(b)* [52], although a significant volume of wider micro- and narrow mesopores also existed. The S_BET_ was measured as 1161 m^2^/g. The above textural characteristics are assumed as beneficial for utilization as a support/composite-filler, in order to enable the dispersion of the photo-active inorganic phase [53,54,55,56] and to promote mass transfer/diffusion phenomena [57]. The XRD pattern of biochar clearly indicated that the material was amorphous and no other crystallinities existed as impurities.

### 3.2. Characterization of BiOCl-Biochar Composites

The first goal of the multifunctional composites’ design and development concept was to determine which amount of biochar can be assumed as the optimum one. Hence, four different loadings of biochar were studied. As can be seen from the XRD patterns (Figure 2), the addition of the organic phase did not affect the crystallization process. On the contrary, the intensity of the diffractions for BiOCl crystals increased while the theoretical amount of the used powder for the tests decreased due to the addition of biochar. This can suggest that bigger crystals were formed as a result of the biochar acting as a substrate for crystallization. Based on the preliminary photocatalytic tests (the most important will be presented herein afterward at the discussion of the kinetic analysis), the composite with a 15% theoretical amount of biochar (15BCPC) presented the best performance and so the experiments were focused on this sample.

The SEM images of the composite (Figure 1c) revealed that the structure and morphology of either biochar or BiOCl nanoparticles did not change, with the former acting as a substrate for the NPs. The uniform dispersion of the nanoparticles can be concluded both based on the SEM images and more importantly from the EDX analysis (Figure 4) where the homogeneous dispersion of Bi and Cl is observable. It should be pointed out that micro-scaled aggregations of BiOCl were found, although few.

The thermogravimetry analysis (TGA) curves in helium of BiOCl, biochar, and 15BCPC are shown in Figure 5. BiOCl was found very stable even up to 700 °C. The thermal profile of biochar shows two significant weight loss phases. The first and big one of around 29.5 wt.%. until 130 °C can be linked to the removal of adsorbed water/humidity [58] or/and volatile substances from the pores. It can be linked to the high surface area and in addition, can suggest great hydrophilicity of the biochar’s surface [59,60]. Considering that the weight loss up to 180 °C is negligible for BiOCl, ~30% for the biochar and ~5.6% for the composite, it can be concluded that the organic porous biochar phase was around 19% per mass. This value is as expected slightly higher than the targeted 15%, due to the differences in the crystallization process. The total weight losses at 800 °C for BiOCl, biochar, and 15BCPC were around 18%, 72, and 32%, respectively.

The surface area of the composite was found to be 126 m^2^/g. If the material was a physical mixture, the expected surface area taking into account the wt.% of biochar (as derived from TGA), would be (19% *×* 1160 + 81% *×* 19 =) 236 m^2^/g. This difference in S_BET_ can be assigned to the blockage of the pores’ entrance/window as well as to possible arising synergistic effects [61,62]. From the isotherms and the pore size distribution (Figure 3b) plots can be assured that the porous network consisted of pores with a wide range of diameters, which is an important feature for adsorption and catalytic applications since the diffusion of the molecules to the active catalytic sites through the network of pores is promoted. In addition, it can be seen that the wide micropores and the small macropores that were expected due to the presence of the biochar cannot be found, a fact that supports the above-mentioned.

The optical features of the sample were evaluated using diffuse reflectance spectroscopy in the electromagnetic range UV-A-visible and the adsorption spectra are shown in Figure 6a. We can observe that the pristine BiOCl presented negligible light adsorption in the visible range of light but strong adsorption from 380 nm and below. The introduction of biochar resulted in a significant enhancement of light adsorption in the visible range, which is a key feature upon utilization of the nanomaterial for ambient solar light applications. This is in agreement with the color of the materials from white for pure BiOCl to light grey for the composite, as was also reported previously in the case of the incorporation of carbonaceous phases towards composite formation [62]. The Tauc plots (Figure 6b) derived using the Kubelka–Munk function were used for the estimation of the band gap (E_g_) energies that were found as 3.43 and 3.30 eV for BiOCl and 15BCPC, respectively. Distinctly, the band gap energy of the sample decreased with the addition of biochar because biochar acts as a sensitizer and creates a delocalized state inside the BiOCl band gap [63].

The point of zero charge (pH_pzc_) was evaluated to determine the surface charge of 15BCPC. At different pH solutions, the interaction of the photocatalyst surface particles with the solvent ions is distinct, giving rise to different charges on the photocatalyst surface. The pH_pzc_ is the point where pH_final_ = pH_initial_, and we observed in Figure 7 that the pH_pzc_ for 15BCPC is 4.09. Therefore, we can infer that the surface of 15BCPC has a positive charge for a pH < 4.09 and a negative charge for pH > 4.09. Moreover, since we are degrading an anionic dye, i.e., MO, adsorption, and photocatalytic degradation of the dye are favorable at a pH < 4.09 due to the electrostatic attraction. The degradation efficiency of the photocatalyst is supposed to decrease for a pH > 4.09 due to repulsive force since both 15BCPC and MO have a negative charge at this pH.

Based on the above-presented physicochemical characterizations, it is of utmost importance to conclude that some unique features of each counterpart were maintained. The first one is the elevated surface area and the micro- and meso-pores network, which is expected to promote the diffusion of the organic pollutants towards the reactive site in order to be converted or/and adsorbed. The high porosity can be linked to the biochar that acted as a support and its presence led to a homogeneous dispersion of the inorganic photo-active phase. What is also important is that the crystallinity and the nanoscale size of the photoactive nanoparticles were maintained when the synthesis took place with the co-presence of the biochar. The derived synergistic effects upon the composite formation and interaction of the inorganic–organic phase were expected to have a positive impact on the bifunctional adsorptive and photocatalytic remediation efficiency. This was expected not only for the narrowing of the E_g_ in the case of the composite, but also for the well-known effect of the carbonaceous phase when it is used as fillers to favor the photoinduced e^−^/h^+^ pairs separation and delocalization within the entire photocatalyst’s mass [62,64,65].

### 3.3. Control Experiment

The main goal for the composite was to present multifunctionality and to be able efficiently to combine the remediation features of the two counterparts against the targeted organic pollutant, meaning for both to promote the photocatalytic decomposition of as well as to show an elevated adsorption performance. From the point of view of remediation tests, the main task was to optimize the removal bi-functional process using RSM studies. It should be pointed out that the pyrolysis temperature was chosen also based on the preliminary optimization tests.

The control experiments were performed in the dark and under UV light to determine the role of photolysis and adsorption in removing the MO, as shown in Figure 8. The photolysis experiments revealed that the MO is stable and resistant to UV light, since only a 6% decomposition was recorded for a solution of an initial concentration of 30 ppm, using 1 g/L photocatalyst, and at a pH of 6.3 (that remained unchanged through the experiment without an acid/base addition). In the dark, BiOCl and 15BCPC resulted in 19% and 82% MO removal (after 150 min). These values are significantly high also considering the textural features of the materials and suggesting that the composite can perform acceptably even in the dark. Upon UV light irradiation, BiOCl and 15BCPC showed 30% and 99% MO removal (after 150 min) indicating that both samples are photo-active. A clear outcome is that the addition of biochar had a dramatically positive impact on the photo-assisted remediation performance, since when using less than 20% per mass, a more than three times higher remediation efficiency was achieved. In general, it can be concluded that 15BCPC possesses excellent adsorption and photocatalytic efficiency due to its porous structure, high surface area, surface properties, and the homogeneous dispersion of the photoactive phase.

### 3.4. Kinetic Studies

The photocatalytic performance of BC-BiOCl towards the degradation of MO under UV light was also examined kinetically. In Figure 9a the photo-assisted removals of BiOCl composites with different amounts of biochar under the same conditions are collected. It can be observed that 20BCPC exhibited the best photocatalytic performance, although 15BCPC showed a very close performance. For both 15BCPC and 20BCPC, the MO removal reached 100% in 120 min, which is significantly higher than the adsorptive and photocatalytic efficiency of pristine BiOCl, 5BCPC, and 10BCPC within a limited time.

To analyze the MO degradation kinetics and quantify the photocatalytic efficiency of the prepared photocatalyst, the rate constant (k) was evaluated by fitting the experimental data into a pseudo-first-order kinetic equation:ln (C_0_/C_t_) = −k_app_t
where C_0_, C_t_, and t are the initial concentration of MO (mg/L), the concentration of MO at time t, and the reaction time, respectively, and k_app_ denotes the apparent rate constant (min^−1^) of the pseudo-first-order reaction. Figure 9b shows the apparent rate constants of Biochar-BiOCl composite photocatalysts, and we can observe that 15BCPC has the best photo-assisted MO removal rate. The k_app_ values of pristine BiOCl, 5BCPC, 10BCPC, 15BCPC, and 20BCPC are 0.00148, 0.0045, 0.01863, 0.02176, and 0.02029 min^−1^ respectively. 15BCPC showed the fastest removal rate, approximately 14.7 times higher than that of pure BiOCl, demonstrating the enhancement in the bi-functional removal ability of samples after introducing biochar. The photocatalytic efficiency of 20BCPC is less than that of 15BCPC because the excessive biochar in the photocatalyst hampers the light absorption capability of pure BiOCl and thus decreases the number of photogenerated carriers.

### 3.5. Statistical Analysis and Optimization Study of MO Removal

To determine the relationship between independent process variables (the catalyst dosage (A), the dye concentration (B), and the pH (C)) and the dependent variable (MO removal) after 60 min of photocatalysis, the results of the experiments designed by BBD were fitted in a second-order polynomial equation. The experiment designed by BBD had a total of 17 experimental runs with three defining factors, as shown in Appendix A. The quadratic polynomial equation, which establishes the relationship between the independent variables and the response using the experimental results, is defined as follows:R_1_ = 100 + 16.66 A − 16.63 B − 7.59 C + 16.37 AB + 6.42 AC − 8.91 BC − 9.99 A^2^ − 8.64 B^2^ − 6.75 C^2^.

In Appendix A, the ANOVA results for the quadratic model are shown. The model F-value of 190.40 and the corresponding *p*-value (<0.0001) imply that the model is significant since there is only a 0.01% chance of having an F-value so substantial that it causes of the noise. A model term is regarded as significant if the *p*-value is less than 0.05 [66]; therefore, for this model, A, B, C, AB, AC, BC, A2, B2, and C2 are significant model terms. Adequate precision, i.e., the signal-to-noise ratio and a ratio greater than 4 [67], are preferable. For this model, the ratio is 44.209 indicating an appropriate signal for moving forward with this model to navigate the design space.

Figure 10 represents the correlation between predicted results and the experimental results of the MO degradation and we can observe that the experimental data fit well into this model. The R^2^ value of 0.9959 indicates a strong correlation between the polynomial model and the experimental results; therefore, we can apply this model to estimate the response for the MO removal within the studied range. The difference between the values of Predicted R^2^ and Adjusted R^2^ is reasonable, i.e., less than 2, and the value of the C.V. is 2.37% which is less than 10%, which reflects the accuracy and the reliability of the experiment.

The impact of the catalyst dosage, initial dye concentration, and pH on the photocatalytic degradation of MO is shown through response surface graphs in Figure 10. The effect of the dye concentration on MO degradation is shown in Figure 10a at a pH of 5 and a reaction time of 60 min. In the Figure, for 0.5 g/L catalyst dosage, MO removal efficiency decreased from 99.3161 to 30 % with the increase in dye concentration from 10 to 50 mg/L; this decrease was almost negligible at higher catalyst dosages such as 1.5 g/L. The reduction in MO removal efficiency was mainly due to the decrease in the number of composite molecules interacting with dye molecules and the decline in the number of photons reaching the photocatalyst caused by the solution’s intensive color [68].

Figure 10b shows the impact of the catalyst dosage on MO removal efficiency at a 30 g/L dye concentration and reaction time of 60 min. We can observe from the graph that the MO removal increased with the increase in the catalyst dosage from 0.5 to 1.1 g/L but decreased gradually with the increase in catalyst dosage up to 1.5 g/L. The rise in catalyst dosage increased the number of active sites available and, thereby, the degradation efficiency; these results are more significant at a higher pH such as 7 (53.68–99.72%). The MO removal efficiency decreased after a 1.1 g/L catalyst dosage due to the agglomeration in the photocatalyst, which interrupted the light passage. A higher catalyst dosage caused high turbidity, leading to increased light scattering and decreasing the number of photons adsorbed by the photocatalyst [69].

Figure 10c shows the effect of the pH on MO removal efficiency at a catalyst dosage of 1 g/L and reaction time of 60 min. We can observe from the curve that at a dye concentration of 50 mg/L, the MO degradation efficiency decreased from 87.17% to 52.09% as the pH changed from 3 to 7. This change in degradation efficiency is negligible at lower concentrations such as 10 mg/L. A change in the electrostatic attraction or repulsion between the photocatalyst and dye molecules caused a decrease in the degradation efficiency of the MO with a change in pH. The photocatalyst surface charge played a role significant in the adsorption of the dye molecules and their dissociation in an aqueous solution. The pHpzc for 15BCPC was determined earlier to be 4.09, which indicates that the photocatalyst surface was positively charged at pH 3–4 and negative at pH 4–7. Therefore, the degradation efficiency of MO, an anionic dye, increases for a pH < 4.09 due to electrostatic attraction and decreases for a pH > 4.09 due to repulsion between the molecules.

### 3.6. Process Optimization and Model Validation

An optimization process was conducted to determine the optimum values of independent variables for MO degradation by the 15BCPC photocatalyst. The independent parameters such as the concentration of dye, photocatalyst amount, and pH were set as “with range,” while the response was set as “maximize” to attain the maximum MO degradation. The optimum conditions for the maximum MO removal (100%) proposed by the model were 1.386 g/L of catalyst dosage, 41.785 mg/L of dye concentration, and a pH of 3.147 after 60 min of photocatalysis. The optimization experiment was done twice to verify the precision of the model, and the average result of these experiments was 100% MO removal. Therefore, this confirms that the BBD model used is reliable and accurate in predicting the MO removal in the studied range.

### 3.7. Reusability Study

The reusability of the 15BCPC photocatalyst for removal of MO was studied at optimum conditions (catalyst dosage 1g/L, dye concentration 30 mg/L, and pH 5) for 180 min up to 3 cycles. After every cycle, the photocatalyst was recovered by centrifuging (for 5 min at 8000 rotations per minute), filtering, and then washing the photocatalyst for up to four consecutive cycles with deionized water and finally kept for drying in an oven at 60 °C. Multiple tests were conducted to overcome the mass loss after each cycle and to gather sufficient samples for the next cycle. The results show that 15BCPC exhibited high photo-assisted removal even after 4 cycles as shown in Figure 11, indicating the photocatalyst′s stability and sustainability. It should be pointed out that the almost 63% removal after 4 cycles can be assumed as a high value, since no regeneration at a high temperature or/and by extraction was followed.

### 3.8. Photo-Remediation Mechanisms

It is well reported in the literature that bismuth oxyhalides (BiOX)-based materials are assumed as an emerging class of photo-active candidates for photocatalytic applications, due to their nano-scaled nature, their layered structure, and above all, their resilient absorption of visible light [70,71]. The main reported active species and mechanisms responsible for BiOX photocatalytic decomposition of MO can be seen in Figure 12. Among the various oxygen-containing reactive species that can be created during photocatalytic processes [72,73] and the photo-induced electrons and holes pairs, the ones that showed to play a key role during the MO decomposition were the holes and the generated ^•^O_2_^−^ radicals, while electrons and hydroxyl radicals (•OH) did not play a direct intensive role [74].

Although, the photocatalytic performance of “pure” bismuth oxyhalide materials was low. Hence, the functionalization/activation, as for instance the formation of oxygen vacancies and defects by chemical treatment/modifications [70,74] or the designing of composites with conductive phases such as graphite or graphitic carbon nitride, in order to enhance the electron delocalization and as a result to prevent the e^−^/h^+^ (charge carries) recombination [51,55,62,70,75,76], can affect positively the photocatalytic environmental remediation efficiency, as it occurred in this study where lignin-derived biochar was used as a filler towards photo-active nano-composites formation.

## 4. Conclusions

In this work, a one-step hydrolysis method was used to synthesize a photo-active composite consisting of lignin-derived biochar and bismuth oxychloride (BiOCl) nanoparticles as bi-functional (photocatalytic and adsorptive) remediation media. The addition of biochar had a positive impact on the removal of the dye MO, with the material having around 20% of biochar to present the best performance. The dimensions of the photo-active platelet/disk-like shaped nanoparticles were 55 to 160 nm in diameter with a thickness of 15–25 nm. The best-performing composite (15BCPC) presented a specific surface area of 126 m^2^/g. Since this value is almost half compared to the expected one in the case of a physical mixture, it is clear that the crystallization of the nanoparticles on the biochar acting as a support, led to the blockage of the pores’ entrance/window. The photocatalytic removal of the MO using 15BCPC followed the pseudo-first-order reaction, and the value of the removal rate constant (K) was found to be about 14.7 times that of pristine BiOCl. BBD was used to design and analyze the experiment and the R2 value of 0.9959 verified that the BBD model was fit to optimize the results. The effect of different process variables was studied using response surface graphs. The optimized conditions for maximum degradation (100%) of the MO were found at 1.386 g/L of the catalyst dosage, a pH of 3.147, and a dye concentration of 41.785. The photocatalyst 15BCPC showed stable photocatalytic efficiency (85%) even after three cycles of reusability.

## Figures and Tables

**Figure 1 nanomaterials-13-00735-f001:**
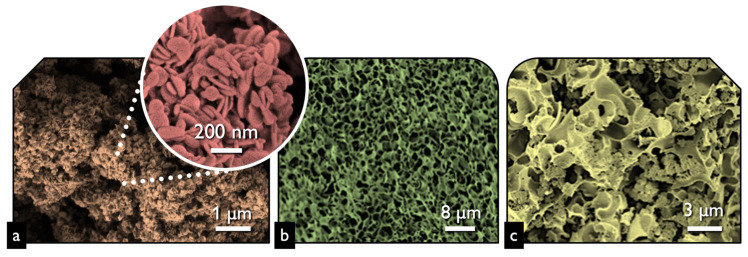
The SEM image of BiOCl (**a**), lignin-derived biochar (**b**), and BiOCl-biochar composite 15BCPC (**c**).

**Figure 2 nanomaterials-13-00735-f002:**
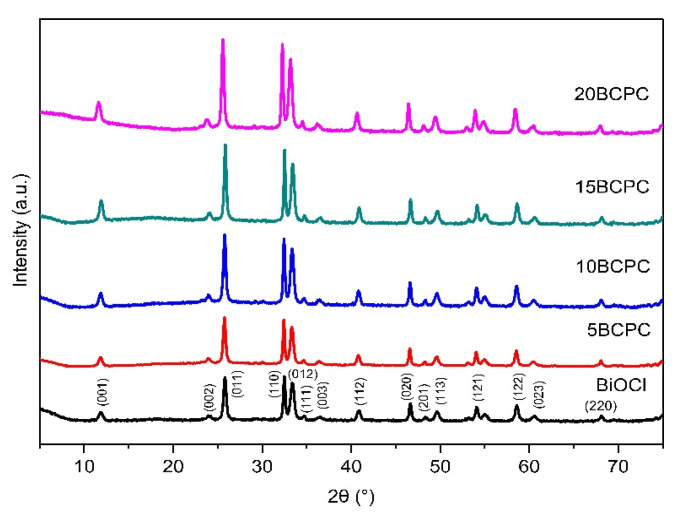
The XRD patterns of pure BiOCl nanoparticles and their composites with different amounts of lignin-derived biochar.

**Figure 3 nanomaterials-13-00735-f003:**
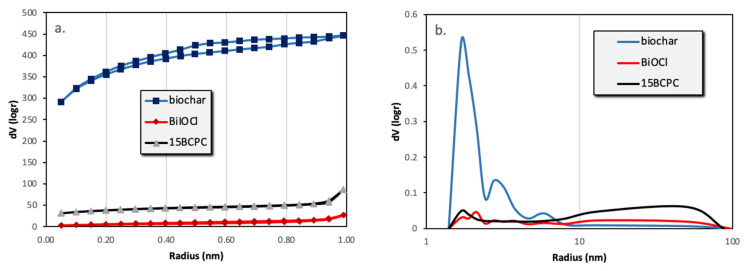
The nitrogen adsorption–desorption isotherms (**a**) and the pore size distributions (**b**) of biochar, BiOCl, and 15BCPC.

**Figure 4 nanomaterials-13-00735-f004:**
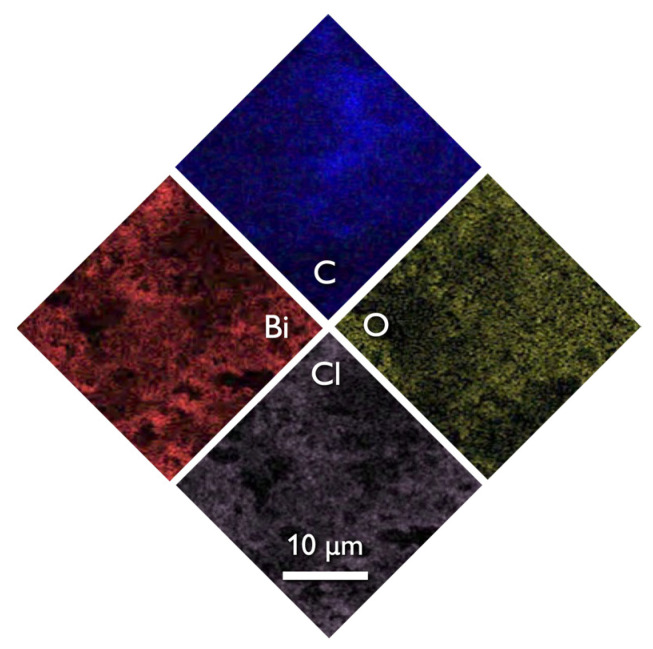
The SEM image of EDX spectroscopy of 15BCPC.

**Figure 5 nanomaterials-13-00735-f005:**
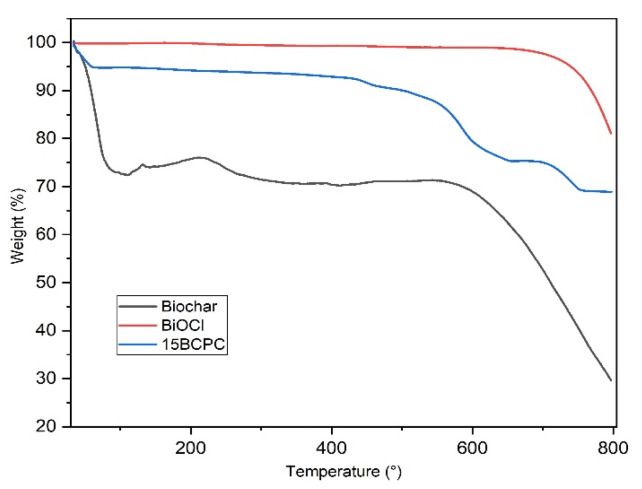
The TGA curves of biochar, BiOCl, and 15BCPC under a nitrogen atmosphere (heating at a rate of 10 °C/min).

**Figure 6 nanomaterials-13-00735-f006:**
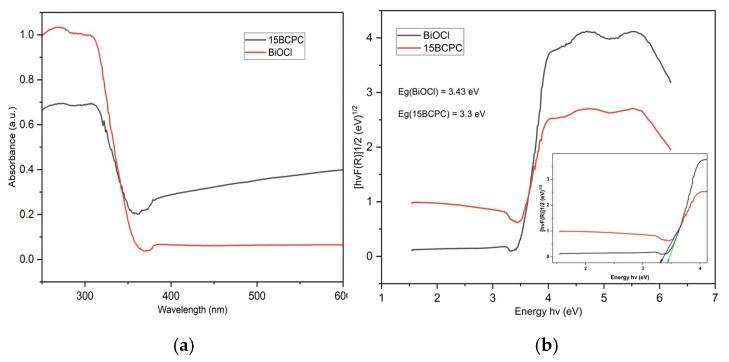
The UV–Vis diffuse Reflectance Spectra of BiOCl and 15BCPC (**a**) and the derived Tauc plots (**b**) for the estimation of the band gap.

**Figure 7 nanomaterials-13-00735-f007:**
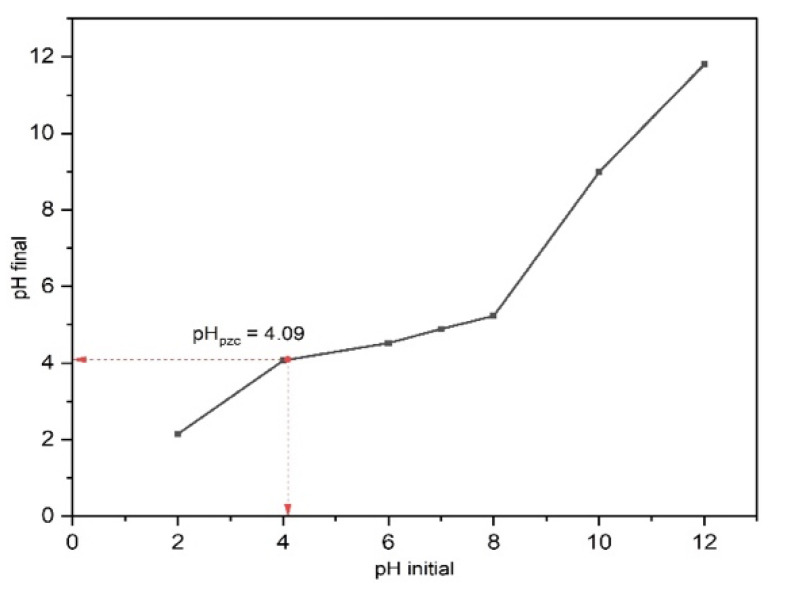
The pH_pzc_ plot of 15BCPC using a 0.1 M NaCl solution.

**Figure 8 nanomaterials-13-00735-f008:**
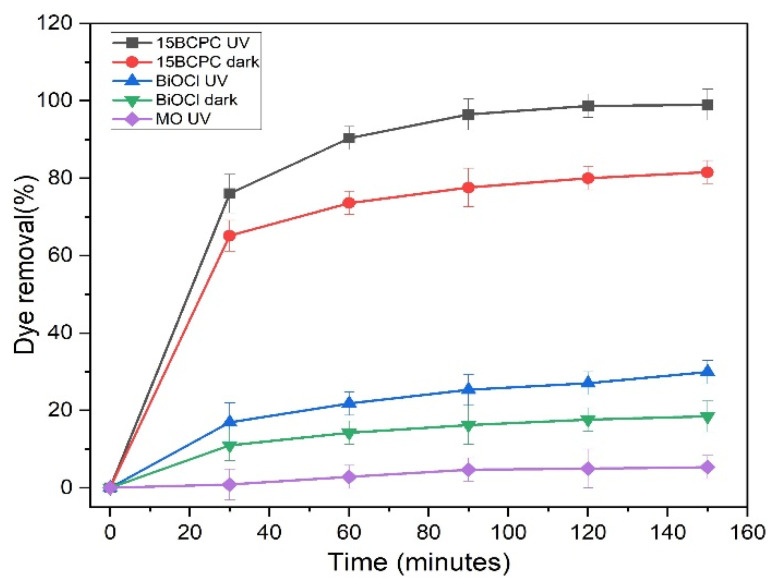
The MO removal under controlled conditions.

**Figure 9 nanomaterials-13-00735-f009:**
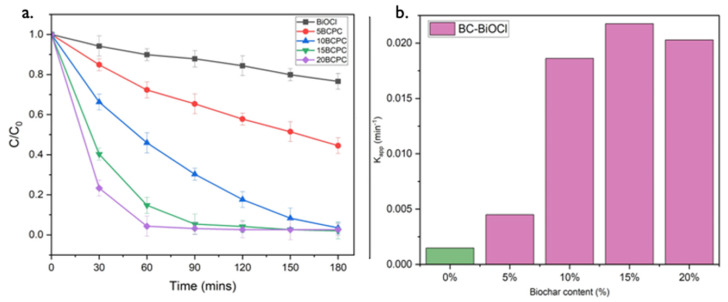
The photo-assisted removal evolution of MO (30 mg/L) using pristine BiOCl (1 g/L) and its composites with different amounts of lignin-derived biochar (**a**) and the apparent rate constants of the MO removal (**b**) under UV light irradiation.

**Figure 10 nanomaterials-13-00735-f010:**
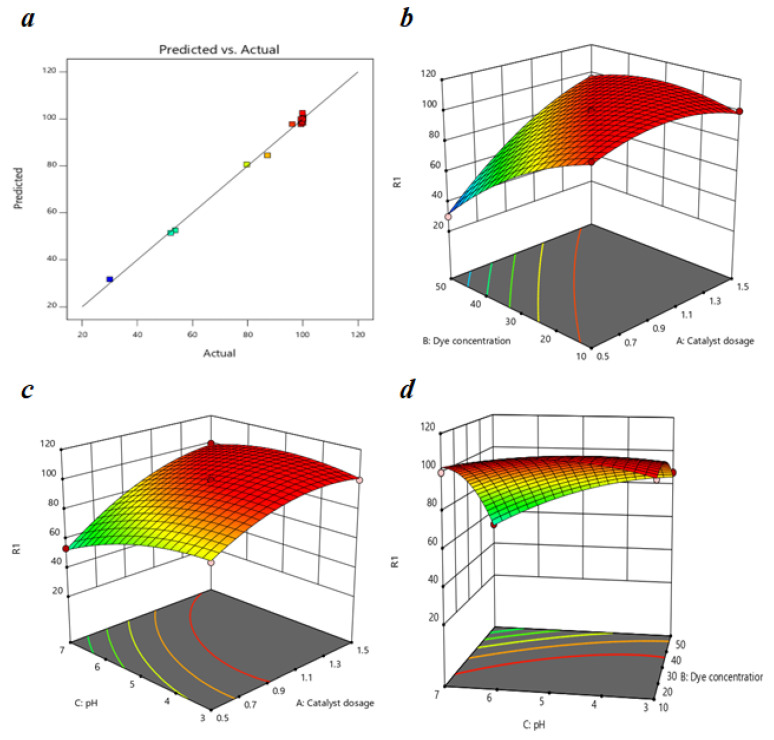
(**a**) A plot of predicted versus actual efficiency for MO removal (%); response surface graphs of MO removal as a function of (**b**) dye concentration and catalyst dosage, (**c**) pH and catalyst dosage, and (**d**) pH and dye concentration respectively.

**Figure 11 nanomaterials-13-00735-f011:**
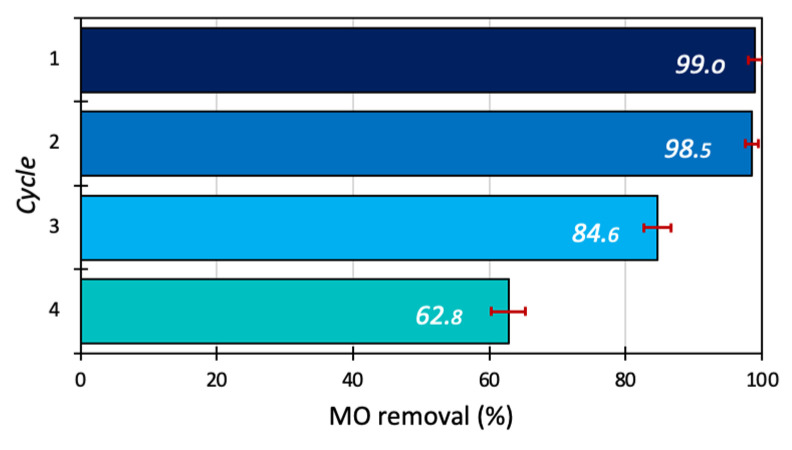
The reusability test of 15BCPC (catalyst dosage 1 g/L, dye concentration 30 mg/L, and pH 5).

**Figure 12 nanomaterials-13-00735-f012:**
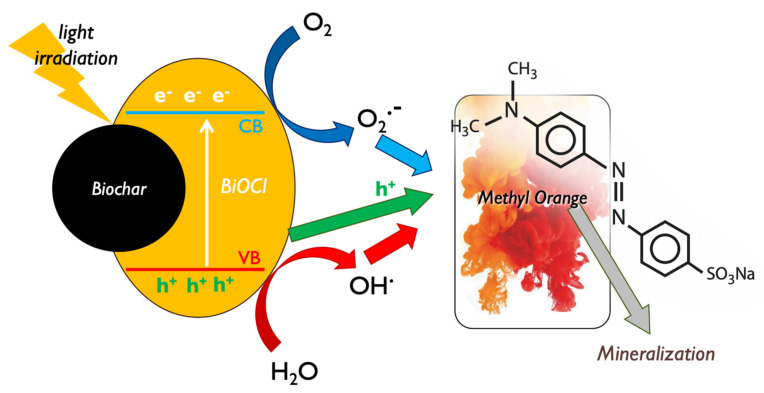
The involved active species during the photocatalytic methyl orange decomposition by bismuth oxychloride composite with lignin-derived biochar.

## Data Availability

Data sharing not applicable.

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
