# Peer review of "Composites of Lignin-Based Biochar with BiOCl for Photocatalytic Water Treatment: RSM Studies for Process Optimization"

_nanomaterials, 2023, doi:10.3390/nano13040735_

Round 1

Reviewer 1 Report

Composites consisting of BiOCl and lignin-based biochar were synthesized following a one-step hydrolysis synthesis. The photocatalytic and adsorptive efficiency of the Biochar-BiOCl composites were studied for the degradation of azo anionic dye, and methyl orange dye. Overall, the research has been able to be the material synthesis process as well as the efficient photocatalysis process. However, this study needs some improvement before it can be accepted.

1. Figure 3b, pore size does not clearly show pore size, the author should re-analyze, remove, or put in the supporting information.

2. Figure 4, SEM and EDX photos are not showing clearly, please check again

3. What is the difference in Figure 6, TGA of the samples? Authors should be careful in providing appropriate data, analysis and mentions.

4. The author can additionally distribute the UV-Vis spectrum of the photocatalytic processes mentioned in the figure.

5. The author should cite the documents below to enrich this study, as well as to better explain the above issues such as TGA, SEM, EDX and UV-Vis results, as well as the mechanism of photocatalysis. .

https://doi.org/10.1016/j.matlet.2021.131129

https://doi.org/10.1007/s11356-021-13597-z

https://doi.org/10.1016/j.ijhydene.2021.11.063

Reviewer 2 Report

1. Review all graphics, subtitles are small, ariel, no pattern. This seems irrelevant but it organizes the work for the reader.

2. The quality of the figures is poor, it should be improved.

3. Please provide the IR data and related discussion, the ref should be updated, such as Inorganics, 10(2022) 202

4. “A significant amount of dye enters the environment through industrial effluents from the textile, food, petrochemical, and pharmaceutical industries’ this part should be updated some refs, such as CrystEngComm, 2022, 24, 6933–6943 and Mater. Today. Commum., 2022, 31,103514.

5. why the author only test 3 recycle, it should be checked the 4 running.

6. One of the most important factors affecting the degradation performance of photocatalysis is air humidity, etc. Why did the author consider the effect of pH?

7. EPR are characterized used to explain the photocatalysis mechanism, there is no EIS and other tests further explain the mechanism

8. There is no photocatalytic mechanism diagram

Reviewer 3 Report

Manuscript deals with the preparation of composites of Lignin-based Biochar with BiOCl for photocatalytic water treatment,  but publication point of view some modification necessary.

1. There are few grammatical mistakes. Please check the manuscript for grammar and English.

2. What is novelty of the present work? Rewrite it at the end of introduction section.

3. Compare your photocatalytic results with the other other researcher work in tabular form.

4.  Recycling experiments should be provided, and XRD or XPS should characterize  composites after photocatalytic reaction.

5. Why author choose MO for degradation?

6.  To enrich literature in the introduction add some references.

i. Nanomaterials 13 (2), (2023), 338 ii. Recent Patents on Nanotechnology 17 (1), (2023), 5-7, iii. Catalysts 12 (10), (2022), 1290, iv. Journal of Alloys and Compounds 928, (2022), 167133

7. Add XPS study to confirm the oxidation state  of prepared materials.

8. Check the references and make it uniform.

Reviewer 4 Report

Nair et al used biochar-BiOCl for dye degradation under UV light irradiation. The effect of biochar on the performance of BiOCl has been well investigated and the influence of various parameters on the degradation efficiency is clarified in this manuscript. This work can provide some useful information to the readership, but some issues should be addressed before accepting this work.

1. The 15%BC-BiOCl exhibits high adsorption towards MO (82%), and the contribution of UV light seems to be low. The author should comment more on this part. Also, please add the control experiment to test the adsorption ability of BC on MO.

2. The reaction mechanism should be supplemented by adding different scavengers to detect the reaction species.

3. The photoelectrochemical analysis, such as photocurrent, EIS, PL et al should be supplemented to probe the charge separation.

4. The relative references should be cited: Nat Commun 9, 1543 (2018); Chemical Engineering Journal 446 (2022) 137316

Round 2

Reviewer 2 Report

it can be accepted.

Reviewer 3 Report

Revision made by the author satisfactory and present form of manuscript should be accepted.